# Attitudes and Knowledge of European Medical Students and Early Graduates about Vaccination and Self-Reported Vaccination Coverage—Multinational Cross-Sectional Survey

**DOI:** 10.3390/ijerph18073595

**Published:** 2021-03-30

**Authors:** Olga M. Rostkowska, Alexandra Peters, Jonas Montvidas, Tudor M. Magdas, Leon Rensen, Wojciech S. Zgliczyński, Magdalena Durlik, Benedikt W. Pelzer

**Affiliations:** 1Department of Transplantation Medicine, Nephrology and Internal Diseases, Medical University of Warsaw, Nowogrodzka 59, 02-006 Warsaw, Poland; magdalena.durlik@wum.edu.pl; 2European Medical Students’ Association (EMSA), Rue Guimard 15, 1040 Brussels, Belgium; alexandra.peters86@googlemail.com (A.P.); jo.montvidas@gmail.com (J.M.); tmmagdas@gmail.com (T.M.M.); leonrensen89@gmail.com (L.R.); 3Department of Surgery, Klinikum Porz am Rhein, Urbacher Weg 19, 51149 Cologne, Germany; 4LUHS Hospital Kaunas, Lithuanian University of Health Sciences, A. Mickevičiaus g. 9, 44307 Kaunas, Lithuania; 5Faculty of Medicine, Iuliu Hatieganu University of Medicine and Pharmacy, Strada Victor Babeș 8, 400000 Cluj-Napoca, Romania; 6Leiden University Medical Center, Faculty of Medicine, University of Leiden, Albinusdreef 2, 2333 ZA Leiden, The Netherlands; 7School of Public Health, Centre of Postgraduate Medical Education, Kleczewska 61/63, 01-826 Warsaw, Poland; wojciech.zgliczynski2@cmkp.edu.pl; 8Department I of Internal Medicine, Center for Integrated Oncology Aachen Bonn Cologne Duesseldorf, University of Cologne, Kerpener Str. 62, D-50931 Cologne, Germany

**Keywords:** medical students, vaccination, vaccination programs, influenza, medical education

## Abstract

Vaccination is one of the most useful preventive interventions in healthcare. The purpose of our study was to gain overview of the opinions, knowledge, and engagement in vaccination practices among medical students (MS) and junior doctors (JD) in Europe. The survey was distributed from March 2016 until August 2016 via the e-mail and social media of the European Medical Students’ Association. In total, 1821 responses from MS and JD from 34 countries in the European region were analysed. The majority of respondents agreed that vaccines are useful (98.7%) and effective (97.2%). Although the necessity of revaccination was supported by 99.2%, only 68.0% of the respondents went through with it. Even though the potential benefit of the flu vaccination seems to be acknowledged by our participants, only 22.1% of MS and JD declared getting the flu shot every or every other season. MS and JD were in favour of specific mandatory vaccination for medical staff (86.0%) and medical students (82.7%). Furthermore, we analysed the self-reported vaccination coverage of our participants regarding 19 vaccines. Of the respondents, 89.5% claimed to provide advice about vaccination to their friends and family. In conclusion, European MS and JD have a very positive attitude towards vaccination. However, their behaviour and knowledge demonstrate certain gaps which should be further addressed in medical education.

## 1. Introduction

Vaccination is one of the most powerful preventive health-interventions leading to improvement of survival rates and the reduction of disease burden [1,2,3]. Nonetheless, even though vaccines undergo rigorous testing, anti-vaccination movements and vaccine scepticism are on the rise [4,5]. The reasons for that phenomenon are diverse: mistrust with governmental institutions, increased media presence of vaccination opponents, obstacles in access to healthcare or limited general knowledge about vaccines [6,7,8]. At the same time, outbreaks of vaccine-preventable diseases show the importance of proper vaccination counselling [9]. This topic gains particular importance in times of the SARS-CoV-2 pandemic and the subsequent development of COVID-19 vaccines [10]. Despite the health threats of coronavirus transmission and its societal impact, a significant proportion of the population remains sceptical about vaccination against COVID-19 [11].

However, since physicians play a key role in vaccination advocacy, the importance of structural training regarding vaccination during medical studies cannot be overstated.

Therefore, several studies explored the knowledge and attitudes of medical students (MS) on vaccination, applying surveys, laboratory approaches combined with questionnaires and educational interventions [12,13,14,15]. Nevertheless, most of these studies included only one or a very limited number of vaccines, e.g., against hepatitis B or seasonal influenza [14,16]. Knowledge, attitudes, and immunisation rates differ markedly between the studies depending on nationality and diseases considered, suggesting the relevance of the socio-cultural background [17]. For instance, roughly half of MS in a Brazilian medical school claimed to be vaccinated against hepatitis B, compared to 60% in a Saudi Arabian and 86% in a German medical school [13,18,19,20,21]. Another example is the Human papillomavirus (HPV) vaccine with one study showing that 65% of female MS from the Czech Republic and only 21% of female MS from Slovakia had been vaccinated against HPV [22]. This reconfirms that the scale of vaccination uptake in MS is very diverse and usually cannot easily be explained within one study. Multiple research projects in this area are necessary, including international and regional ones, to explore the drivers behind vaccination practices and attitudes of MS in Europe and globally.

The purpose of our study was to gain a better understanding and overview of the opinions, knowledge, and engagement in vaccination schedules among MS and junior doctors (JD) in Europe in order to identify areas of incertitude and indicate potential gaps in medical education. It was also aimed to determine whether MS and JD are aware of their own vaccination status for 19 specific vaccine types.

## 2. Materials and Methods

### 2.1. Questionnaire Description

The survey (Appendix A) contained 31 questions (Q); 30 single-choice closed questions and one open-question. Single-choice questions were presented as A-B-C-D where students were asked to select one option. All questions except Q24 (additional vaccination) were obligatory for completion of the form.

All question items investigated 7 areas: general attitude towards vaccination and sources of information (Q1-3), knowledge and practices regarding vaccination boosters (Q4, Q25), vaccination counselling and vaccination programmes in pregnancy (Q26, Q27), seasonal influenza vaccination (Q28, Q29), mandatory vaccination of medical staff and medical students (Q30, Q31), self-reported coverage for 19 selected vaccines (Q5-Q24) and additional vaccination (Q24). The questionnaire began with open questions on the year of birth, country of enrolment for studies, studied subject and the year of studies. A closed question on the gender offered three choice options: female, male or other. If the participants did not identify themselves as either female or male, or did not feel like disclosing it, they could select ‘other gender’. The survey was not based on any previously available tool and the questions were created for the research purpose by an international team of medical students involved with the European Medical Students’ Association (EMSA Europe). The study was piloted and adapted within the research team and the board of EMSA Europe, but the responses were not included in the final analyses. The questionnaire was granted ethical approval by the Executive Board of EMSA Europe after a careful evaluation of its anonymity and dissemination methods. By filling the questionnaire, participants agreed to contribute to the research project, thus no individual consent forms were collected. Consent was assumed from people who took part in the study and recorded their responses in the survey system. The study was anonymous, and no incentives were offered in exchange for participation.

### 2.2. Dissemination Methods

Messages including a description of the project and the questionnaire were sent out through the e-mail network of EMSA Europe. EMSA Europe’s representatives disseminated it further at their local universities. It was also announced via e-mails and on social media in various medical student groups known to the authors. The number of respondents per country varied due to factors such as different ways the survey was disseminated locally and how actively was it promoted by the students involved. Additional measures were taken such as sending out reminders or reaching out to individuals from under-represented countries as well as study years within a country with lower participation. Survey responses were collected from March 2016 until August 2016. The response rate was non-calculable in the dissemination method applied.

### 2.3. Statistical Methods

For the analysis, we used Microsoft Excel and SPSS version 25 software (IBM, Armonk, NY, USA). We grouped medical students and graduates according to their year of study: 1–2, 3–4, 5–6 or graduates. Three students studied in years 7 or 8 and they were aggregated with respondents from year 5–6. Seven individuals declared their gender as ‘other’ and they were excluded from the comparison of answers provided by males and females, but they were included in the year-of-study and overall analysis. All the analysed variables were categorical. Counts (*n*) and percentages (%) were estimated for answers provided in every question by from the total sample, by males, by females, by students from years, 3–4 and 5–6 years of study and junior doctors. Due to expected counts smaller than 5 in some cells in the contingency table, Fisher’s exact test was used to compare answers provided by males and females as well as to compare answers provided by students from years, 3–4 and 5–6 of study and graduates. Statistical significance was based on the criterion *p* < 0.05.

## 3. Results

In total, we collected 1978 responses. We excluded 27 known duplicates or records with incorrect socio-geographic information and 119 respondents who studied subjects other than medicine. Eleven MS were excluded as they lacked a comprehensive understanding of the concept of vaccinations. Information provided by the remaining 1821 MS and JD was further analysed.

The mean age of 1821 MS and JD was 23.6 years (range: 18 to 44 years, Standard Deviation 2.75 years). The characteristics of MS and JD are presented in Table 1 and Figure 1. Three respondents in years 7. and 8. were grouped with MS in years 5–6 A database of the analysed responses from the survey is available in the Appendix A.

### 3.1. General Attitude towards Vaccination and Sources of Information

Identifying the overall opinion of MS and JD about vaccination, 98.7% respondents declared that ‘vaccines are useful, and everybody should be vaccinated’. We found that year MS agreed less often (96.6%, *p* < 0.001). Regarding the impact of vaccination programs, 97.2% of MS and JD were convinced that they are effective with lower scores among year students (93.8%, *p* < 0.001). Considering factors influencing the opinion about vaccination, 88.4% of MS and JD chose ‘scientific facts’. No respondent chose ‘Religious beliefs’ as their source of knowledge. There were no statistically significant differences between females and males in those questions (Table 2).

### 3.2. Knowledge and Practices Regarding Booster Vaccination

Roughly 97% of respondents declared being vaccinated as part of the national vaccination program in their country. Interestingly, 1–2 year MS declared being vaccinated less often in the national vaccination programs (93.8%, *p* = 0.004) than students in the higher grades and JD. While the awareness of booster vaccination in general was shown by nearly all MS and JD (99.2%), performing it was declared by only 68.0% of respondents, again showing lower scores in 1–2 year MS (77.6% vs. 59.7%, *p* < 0.001). There were no statistically significant differences between females and males in those questions (Table 3).

### 3.3. Vaccination Counselling and Attitudes towards Vaccination Programs in Pregnancy

In terms of advising friends, family and colleagues about vaccination, 89.5% of respondents declared doing so, with females providing counselling more often (91.0% vs. 85.9%, *p* = 0.005). We found that 5–6 year MS and JD were more active in vaccine counselling than the total studied population in our survey (95.5% and 94.8%, respectively, *p* < 0.001). Concerning specific vaccination programmes being available to pregnant women, over 82.8% of respondents were in favour of such solutions. This option was less frequently chosen by 1–2. year MS (76.7%, *p* < 0.001) and tended to be lower in females as well (81.4%, *p* = 0.085) (Table 4).

### 3.4. Attitudes towards Seasonal Influenza Vaccination

About 75.7% of MS and JD declared that even though the seasonal flu vaccine may not prevent an individual from becoming ill, the symptoms would be milder, showing a significant difference regarding stage of medical education (87.9% JD vs. 66.0% 1–2 year MS, *p* < 0.001). Interestingly, males more often believed that a vaccine prevents flu completely (9.7% vs. 5.7%, *p* = 0.020). Even though the benefit of the flu vaccination seems to be acknowledged by our participants, only 22.1% of MS and JD declared getting the flu shot every or every other season. Females more often declared they never got the flu vaccine (47.6% vs. 42.2%, *p* = 0.022) and more 1–2 year MS alike compared to 5–6 year MS (49.9% vs. 41.6%, *p* < 0.001) (Table 5).

### 3.5. Attitudes towards Mandatory Vaccination of Medical Staff and Medical Students

Most individuals agreed that mandatory vaccination programs for medical staff against influenza and hepatitis B should be in place (86%). When asked about identical programs dedicated to MS, 82.7% participants declared support. Students in clinical rotations were more confident about mandatory vaccination for medical students—84.0% and 87.3% for 3–4 and 5–6 year MS, respectively, compared to 74.7% in 1–2 year MS (*p* < 0.001). There were no statistically significant differences between females and males in those questions (Table 6).

### 3.6. Self-Reported Coverage for 19 Selected Vaccines

MS and JD reported coverage for 19 selected vaccines in country-specific vaccination programmes (Table 7 and Table 8). The vaccines were divided into four categories: childhood schedules in all EU/EEA, additional childhood vaccines, vaccines recommended both for children and for adults in risk groups, vaccines for adults. The division was based on recommendations of the European Centre for Disease Control and Prevention (ECDC) and the European Commission [23]. Data is presented in Table 7—total sample and against gender, and Table 8—against year of study.

The most frequently declared vaccine was tetanus (94.2%) and least was cholera (12.9%) (Table 7).

Females more often declared being vaccinated against rubella, HPV, hepatitis B and typhus (*p* < 0.05). Males prevailed in being vaccinated against meningococci, seasonal influenza and cholera (*p* < 0.05). Vaccine against HPV presented the biggest differences between female (33.1%) and male (2.7%) participants (*p* < 0.001), (Table 7).

MS from years 5–6 and JD more often declared being vaccinated against tetanus, diphtheria, poliomyelitis, rubella, measles, mumps, pertussis, *Haemophilus influenzae* type b, tuberculosis, chickenpox, hepatitis B and seasonal influenza (*p* < 0.001). Contrary to that, 1–2 year MS more often reported on being vaccinated against HPV, hepatitis A, meningococci, pneumococci, typhus and cholera (*p* < 0.001). In case of all 19 vaccines, 1–2 year MS were least certain of their vaccination status and ‘did not know’ how to answer most often among the year groups (*p* < 0.001) (Table 8).

### 3.7. Additional Vaccination

In our study, 370 MS and JD declared that they had been additionally vaccinated against communicable diseases different than mentioned in the previous questions. Six other vaccines were identified: against tick-borne encephalitis (*n* = 218, 58.9%), yellow fever (*n* = 83, 22.4%), rabies (*n* = 42, 11.4%), Japanese encephalitis (*n* = 17, 4.6%), Swine flu (*n* = 6, 1.6%) and rotavirus (*n* = 4, 1.1%).

## 4. Discussion

The outcomes of our survey demonstrate that the majority of MS and JD believe that vaccines are a safe and a useful tool in prevention of infectious diseases. Nearly all the participants declared their support for vaccines, including vaccination programs which resonates with findings of other authors [24,25,26]. However, geographical and socio-cultural disparities may limit direct comparability of our results [17]. Nonetheless, our study revealed certain knowledge gaps, inconsistencies and discrepancies between different groups of MS and JD considering vaccination.

As a recurrent finding, vaccination coverage and knowledge were in general lowest in MS in their 1–2 years of study and increased gradually with the consecutive years of medical school. Awareness and theory consolidate along with the university and hospital training, which was already noticed by Banaszkiewicz et al. who studied this topic in Polish MS [27]. A significant increase of knowledge on immunisation from second (55%) to fourth year (74%) of medical school was also noticed by Berera and Thompson in American students [25]. Despite different socio-geographic characteristics of students, these results are similar to our findings where MS in higher grades of medical school and JD were more confident in their responses and supportive towards vaccination. Efforts should be made to further promote this process. To catalyse knowledge acquisition and make it more attractive, e.g., summer camps on vaccinology as described by Vorsters et al. could be an interesting complement to the academic curriculum where students would learn in a non-academic setting and in a peer-to-peer manner [28,29]. Universities or student groups might consider organising such events to unload the core medical program while allowing more practical training, for instance on vaccine counselling.

The most significant source of opinions on vaccination chosen in our study were ‘scientific facts’ (88.4%), although the answer does not specify where those facts are obtained from. The Internet definitely plays an increasing role in medical knowledge formation as 1.9% of 1–2 year MS chose social media compared to 0.0% JD. A German study published in 2012 identified that 63.5% of medical trainees see the Internet as a relevant source of health information, including vaccination [30]. Although comparison with groups outside Europe may be biased, similar results were found among 4th year medical students of Lahore Medical & Dental College in Pakistan [31]. The conclusions of those findings should be embraced by medical schools as e-learning tools, including social media engagement, have become strongly intertwined with modern medical education.

Only 68.0% of MS and JD stated that vaccination boosters are important for maintaining protection against certain diseases and declared compliance with ‘reminder’ immunisation schedules. The WHO recommends vaccination boosters for healthcare workers for a number of vaccines, including diphtheria, meningococci and seasonal influenza [32]. Data is scarce on how these recommendations are followed among MS and JD. However, our findings demonstrate that efforts to improve rates of vaccination boosters coverage among medical trainees should be in place.

Concerning vaccination programs for pregnant women, most participants were in favour by expressing 82.8% support. It should be underlined that even in pregnancy certain vaccines are not only safe but necessary as the burden of potential diseases outbalances the risks of recommended immunisation. Reinforcing message about preventive immunisation for women in gestation is justified by several studies and recommended by the WHO and the Centers for Disease Control and Prevention (CDC) [33,34,35]. Students see the importance of such interventions, although 11.4% of MS and JD still doubt safety or effectiveness of those programmes. Female participants tended to express slightly less support for such initiatives (81.4% females vs. 86.1% males, *p* = 0.085). This small discrepancy can be due to the bias of decision-making depending on the role, examined by Zikmund-Fisher et al. [36]. The authors of the study analysed in simulated scenarios individual reactions to different medical procedures, including flu vaccination, and concluded that participants making treatment choices for themselves were more apprehensive, at the same time, showing more confidence when deciding for others. Female students would be recipients of pregnancy-intended vaccines therefore exposed to any potential side effects or logistical inconveniences. Furthermore, corresponding literature shows that women in the antenatal period show certain scepticism towards additional vaccination [37,38]. Recalled studies refer to American and Australian populations therefore transferability remains in question. However, this trend might be applicable also to European female MS in our study, even though the statistical difference with males was not significant (*p* = 0.085). Nonetheless, education about indications for vaccination before or during pregnancy should be further emphasised during routine classes for MS and JDs.

The majority of the respondents stated that vaccination against seasonal influenza and hepatitis B, should be mandatory for medical staff and MS. However, support for this vaccination scheme in MS and JD was 3.3%-points lower compared to the option concerning medical staff (82.7% vs. 86.0%). One possible explanation is that for our participants the term ‘medical staff’ is more abstract, while ‘medical student’ is day-by-day reality which remains in line with the findings of Zikmund-Fisher et al. on the matter of perspective when making treatment decisions [36]. Regardless of the high rate of declared support for mandatory vaccination, 45.9% of MS and JD in our study had never been vaccinated against seasonal influenza and as few as 18.1% get vaccinated every year as recommended [39,40]. This is worrying and corresponds with findings of Lorenc et al. that healthcare institutions struggle to get a satisfactory coverage among their staff for seasonal influenza vaccine [41]. Such reluctance may stem from concerns about adverse reactions, logistical challenges in delivering the intervention, its individual costs as well as the perception that influenza is not a harmful disease, especially in young people [41,42,43]. The WHO and CDC continuously advocate for regular immunisation against seasonal influenza in HCW, including MS [39,40]. Providing opportunities for on-campus vaccination and making it free of charge for MS could help establish correct habits and increase immunisation rates among students. Afonso et al. demonstrated at Oakland University, US, that integrated curricular interventions such as theoretical education combined with teaching administration technique and on-site peer-to-peer vaccination can help increase the flu shot coverage among medical learners [12]. Such methods would require evaluation in the European populations before drawing solid conclusions. Surely, as future physicians and experts to-be in various medical fields, MS constitute fundaments for preserving both individual and population health. Moreover, they are examples to others. High quality training on vaccines, vaccine-preventable diseases and addressing vaccine hesitancy must therefore become essential part of medical education—especially at the time of pandemic crisis.

For HPV vaccination, rates were highest among participants from years 1–2 being the youngest group, which may reflect changing national policies for the most recent vaccine in our questionnaire. Many European countries have already adopted HPV vaccination into their national schedules and provide it for free to teenagers, including males [44]. The number of male students vaccinated against HPV in our study was significantly lower than females (33.0% vs. 7.2%, *p* < 0.001), likely because the national programs from a decade ago were most often addressed to teenage girls only or included teenage boys at a later stage [44]. Irrespectively, our 7.2% vaccination rate among young European males based on responses collected in 2016 seems relatively high. In comparison, based on a study by Reiter et al. performed in the US in 2010 the number of vaccinated boys was approximately 2% [45], although vaccination policies in the US vary from those in Europe. However, in a sample of 31 Swedish adolescent males aged 16–19 interviewed in 2017 by Grandahl et al., none was vaccinated against HPV (0%) [46]. The overrepresentation of vaccinated males in our study group compared to other samples may be due to the fact that medical students were most likely vaccinated by parents with a medical background and/or higher education themselves [47,48] who, according to a report of the European Commission, have higher confidence in vaccines compared to the population average [49]. Despite declared vaccination status discrepancies between males and females, since HPV vaccine proves effective in preventing genitourinary cancer also in boys [50,51], the differences should be diminished by vaccination programs inclusive of all teenagers, regardless of gender.

Interestingly, for a number of specific vaccines, namely cholera, typhus, HPV, hepatitis A, meningococci and pneumococci, 1–2 year students declared being vaccinated more often when compared with other groups (*p* < 0.001). Vaccines against HPV, meningococci and pneumococci are relatively new in the national vaccination policies and recommendations in Europe [52,53,54] which can explain this predominance. On the other hand, vaccines against hepatitis A, cholera and typhoid are particularly recommended for individuals travelling to locations with inadequate sanitation and unhygienic food processing methods [55,56,57]. The prevalence of vaccination uptake for those three diseases in the group of junior students could be linked to the unprecedented extent of intercontinental mobility among the European youth [58,59]. However, especially vaccination against hepatitis A is also highly relevant at sites of outbreaks and for several groups at risk [60]. Underlying reasons for such findings lay outside the scope of our research and would require further investigation.

In the open section of our survey, 11 respondents mentioned Lyme disease, hepatitis C, malaria, cytomegalovirus infection, anti-dust allergy and West Nile Fever as diseases against which they were vaccinated. At the point of running the survey, none of those vaccines were available thus those students were excluded from further analyses. Such finding indicate that some MS may have difficulties distinguishing between diseases preventable by vaccines and those which are not and might struggle with the whole concept of vaccination. American and European scientific literature provides reports about certain knowledge gaps and disorientation concerning vaccines among healthcare students, including lack of understanding of the mechanism of action, not knowing indications/contraindications and confusing bacterial/viral immunisation [24,26,61]. Efforts to provide top-notch medical education for health-care professionals as future vaccine counsellors should not decelerate in addressing those knowledge gaps.

Interestingly, over 24.3% of MS stated to be vaccinated against smallpox, while the disease was declared to be eradicated by the World Health Assembly in 1980 and ceased to be recommended shortly after [62], several years before vast majority of participants of our study were born. This suggests that self-reporting about vaccination status from MS/JD may not always reflect reality. Researchers Rolnick et al. [63] and Loulergue et al. [15] demonstrated earlier that inconsistencies between self-reporting and real vaccination status can be significant due to no interest in reporting, lack of knowledge, forgetfulness and no vaccination record available at hand. It would be beneficial if MS were encouraged to explore their own immunisation history as part of self-education.

The topic of vaccination in medical personnel and students is being debated in the context of current SARS-CoV-2 pandemic. A study released in January 2021 by Verger et al. showed a significant proportion of healthcare workers in France and French speaking parts of Belgium and Canada were not willing to be vaccinated against COVID-19 [64]. An earlier study, published in October 2020, anticipated scepticism towards COVID-19 vaccine in Maltese medical staff but demonstrated positive approach in those who take seasonal influenza vaccine [65]. Regarding undergraduates, corresponding results were presented by Italian researchers who report that 1 out of 10 university students including healthcare subjects expressed doubts or would not vaccinate against novel coronavirus [66]. Similar findings were demonstrated in the non-European MS. In the US, Lucia et al. found that 23% of MS were unwilling to take a COVID-19 vaccine immediately upon approval of the Food and Drug Administration [67]. In an analysis of Egyptian MS, 46% were hesitant about immunisation against SARS-CoV-2 [68]. These studies illustrate that a direct global threat such as the coronavirus pandemic which started in 2019 does not necessarily translate into mass vaccination uptake among healthcare professionals and undergraduates as of second half of 2020.

Even though our data was collected in 2016 and could not reflect on the COVID-19 vaccination, it echoes how unintuitive opinions and actions can be regarding vaccination in healthcare undergraduates. More studies are needed to examine factors behind the complexity of vaccination hesitancy and non-compliance in future medical professionals.

### Limitations

We observed an unbalanced representation of some countries in the study (Figure 1) which may have introduced bias in response analysis and drawing generalised conclusions for whole Europe. For this reason, we did not perform the evaluation by country or geographical region but focused on gender and the year of study of the participants. The analysis by nation was also problematic due to disproportion in the study years within given countries. Based on our findings, we know that the level of advancement in medical school correlates strongly with given responses thus making per-country analysis questionable.

The response rate in our survey was not calculated due to the dissemination method of the questionnaire, i.e., using social media and e-mail networks. Due to that, it may raise questions about bias and the representativeness of the sample since in theory anybody on the internet could have filled the form. Nevertheless, considering that completing the questionnaire required certain effort, the questionnaire tool was cross shared over various on-line channels but mostly in closed groups of MS and JD or through direct contact, we believe that our survey provided enough record to overcome this bias.

Another issue is the language barrier which might have played a role in filling the questionnaire. Names of certain vaccines in English might sound unfamiliar to some students or could have been confused with other vaccines, e.g. misinterpreting different types of pox. On the other hand, MS and JD find it easy to navigate for information on-line and could easily translate names of given vaccines when uncertain.

Lastly, about 25% of participants declared immunisation against diseases which are not preventable by vaccination to date or are no longer recommended due to eradication of the pathogen, e.g. hepatitis C or smallpox. Awareness of participant-related factors such as language problems in interpreting specialised medical terms in English, inadequate knowledge about immunisation or a lack of focus in filling the questionnaire oblige us to recommend caution in interpreting the presented data.

## 5. Conclusions

European MS and JD strongly agree about vaccines safety and efficacy, and scepticism in this group is marginal. Students often serve as vaccine counsellors to their friends and family, females more often than males. Knowledge and confidence regarding vaccines improves significantly in several aspects in the consecutive years of medical school and most medical students declare support for specific mandatory vaccination programs dedicated to themselves and medical staff. Despite solid awareness and overall support for vaccination against seasonal influenza among trainees, the coverage in this group remains low—nearly half of MS and JD never got the flu vaccine. There are also certain gaps in the knowledge and practice of vaccination in doctors-to-be which call for addressing in medical education and continuous medical training.

## Figures and Tables

**Figure 1 ijerph-18-03595-f001:**
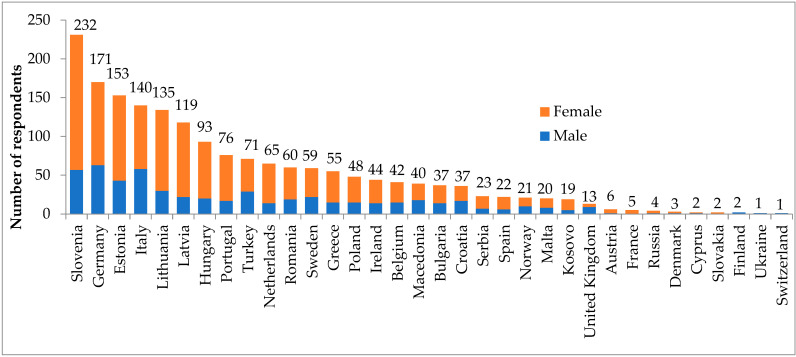
Number of medical students and junior doctors per country (*n* = 1821).

**Table 1 ijerph-18-03595-t001:** General characteristics of the medical students and junior doctors (JD) (*n* = 1821). * including three respondents in years 7–8.

Characteristics	Category	*n* (%)
Year of birth	up to 1991	571 (31.4)
	1992–1993	604 (33.2)
	1994 or higher	646 (35.4)
Gender	Male	555 (30.5)
	Female	1259 (69.1)
	Other	7 (0.4)
Year of study	1–2.	467 (25.6)
	3–4.	645 (35.5)
	5–6. *	651 (35.7)
	JD	58 (3.2)

**Table 2 ijerph-18-03595-t002:** General attitude towards vaccination and sources of information in Q1–Q3 (*n* = 1821).

Question Answer	Total (*n* = 1821)	Gender		Year of Study			
		Male (*n* = 555)	Female (*n* = 1259)	1–2 (*n* = 467)	3–4 (*n* = 645)	5–6 (*n* = 651)	JD (*n* = 58)
Q1. Which statement describes your opinion about vaccinations the most?							
A. It is useful and safe and I think that everybody should get vaccinated.	1797 (98.7)	549 (98.9)	1241 (98.6)	451 (96.6)	640 (99.2)	649 (99.7)	57 (98.3)
B. There is too little evidence to prove that it is effective.	16 (0.9)	4 (0.7)	12 (1.0)	12 (2.6)	3 (0.5)	0 (0.0)	1 (1.7)
C. There is too little evidence to prove that it is even safe to get vaccinated and I think that nobody should do this.	8 (0.4)	2 (0.4)	6 (0.5)	4 (0.9)	2 (0.3)	2 (0.3)	0 (0.0)
Q2. Do you think that vaccination programs are an effective tool in disease prevention?							
A. Yes, I think it is effective.	1770 (97.2)	545 (98.2)	1219 (96.8)	438 (93.8)	630 (97.7)	645 (99.1)	57 (98.3)
B. I don’t think that it makes a difference because I would choose to vaccinate either way.	35 (1.9)	8 (1.4)	26 (2.1)	18 (3.9)	13 (2.0)	4 (0.6)	0 (0.0)
C. No because there is not enough proof that vaccines are effective or even safe.	6 (0.3)	2 (0.4)	4 (0.3)	2 (0.4)	2 (0.3)	2 (0.3)	0 (0.0)
D. No because I don’t think that such things should be forced on.	10 (0.5)	0 (0.0)	10 (0.8)	9 (1.9)	0 (0.0)	0 (0.0)	1 (1.7)
Q3: What influences your opinion about vaccinations the most?							
A. Scientific facts	1610 (88.4)	506 (91.2)	1099 (87.3)	388 (83.1)	573 (88.8)	593 (91.1)	56 (96.6)
B. Social Media	19 (1.0)	4 (0.7)	15 (1.2)	9 (1.9)	3 (0.5)	7 (1.1)	0 (0.0)
C. Senior physicians, professors	159 (8.7)	35 (6.3)	122 (9.7)	48 (10.3)	66 (10.2)	44 (6.8)	1 (1.7)
D. My relatives	26 (1.4)	7 (1.3)	19 (1.5)	20 (4.3)	2 (0.3)	3 (0.5)	1 (1.7)
E. Religious beliefs	0 (0.0)	0 (0.0)	0 (0.0)	0 (0.0)	0 (0.0)	0 (0.0)	0 (0.0)
F. My friends, colleagues	7 (0.4)	3 (0.5)	4 (0.3)	2 (0.4)	1 (0.2)	4 (0.6)	0 (0.0)

Results are presented as *n* (%). Q1 against gender *p* = 0.936, Q1 against year of study *p* < 0.001, Q2 against gender *p* = 0.112, Q2 against year of study *p* < 0.001, Q3 against gender *p* = 0.110, Q3 against year of study *p <* 0.001, *p* for Fisher’s exact test. JD—junior doctors.

**Table 3 ijerph-18-03595-t003:** Knowledge and practices regarding vaccination boosters in Q4 and Q25 (*n* = 1821).

Question Answer	Total (*n* = 1821)	Gender		Year of Study			
		Male (*n* = 555)	Female (*n* = 1259)	1–2(*n* = 467)	3–4 (*n* = 645)	5–6 (*n* = 651)	JD (*n* = 58)
Q4. Did you get vaccinated as part of the vaccination program of your country?							
A. Yes	1765 (96.9)	537 (96.8)	1221 (97.0)	438 (93.8)	629 (97.5)	640 (98.3)	58 (100)
B. No	25 (1.4)	6 (1.1)	19 (1.5)	13 (2.8)	7 (1.1)	5 (0.8)	0 (0.0)
C. I don’t know	31 (1.7)	12 (2.2)	19 (1.5)	16 (3.4)	9 (1.4)	6 (0.9)	0 (0.0)
Q25. Do you know that in order to be protected properly you need to get revaccinated for several vaccines?							
A. Yes, I am aware it and doing it properly.	1239 (68.0)	375 (67.6)	860 (68.3)	279 (59.7)	420 (65.1)	495 (76.0)	45 (77.6)
B. Yes, I am aware of it, but I am not sure if I have full vaccination.	568 (31.2)	173 (31.2)	392 (31.1)	184 (39.4)	220 (34.1)	151 (23.2)	13 (22.4)
C. No, this is the first time I hear about that.	12 (0.7)	6 (1.1)	6 (0.5)	4 (0.9)	5 (0.8)	3 (0.5)	0 (0.0)
D. No, there is no need because vaccination is always life-long protection	2 (0.1)	1 (0.2)	1 (0.1)	0 (0.0)	0 (0.0)	2 (0.3)	0 (0.0)

Results are presented as *n* (%). Q4 against gender *p* = 0.481, Q4 against year of study *p* = 0.004, Q25 against gender *p* = 0.385, Q25 against year of study *p* < 0.001, *p* for Fisher’s exact test. JD—junior doctors.

**Table 4 ijerph-18-03595-t004:** Vaccination counselling and attitudes towards vaccination programs in pregnancy in Q26 and Q27 (*n* = 1821).

Question Answer	Total (*n* = 1821)	Gender		Year of Study			
		Male (*n* = 555)	Female (*n* = 1259)	1–2(*n* = 467)	3–4 (*n* = 645)	5–6 (*n* = 651)	JD (*n* = 58)
Q26. Do you advise your relatives, friends, colleagues etc. to get vaccinated?							
A. Yes	1629 (89.5)	477 (85.9)	1146 (91.0)	366 (78.4)	586 (90.9)	622 (95.5)	55 (94.8)
B. No	52 (2.9)	22 (4.0)	30 (2.4)	26 (5.6)	11 (1.7)	13 (2.0)	2 (3.4)
C. Never thought about that	140 (7.7)	56 (10.1)	83 (6.6)	75 (16.1)	48 (7.4)	16 (2.5)	1 (1.7)
Q27. Do you think that a more specific vaccination program should be available to pregnant women (e.g., seasonal flu, mumps, rubella)?							
A. Yes, because that way the fetus will be protected against inborn anomalies and fewer miscarriages will occur	1507 (82.8)	478 (86.1)	1025 (81.4)	358 (76.7)	536 (83.1)	565 (86.8)	48 (82.8)
B. No, because a specific programme is not safe to a pregnant woman or the fetus	194 (10.7)	46 (8.3)	145 (11.5)	67 (14.3)	70 (10.9)	55 (8.4)	2 (3.4)
C. No, because the vaccine is not effective for the pregnant woman or the fetus.	12 (0.7)	4 (0.7)	8 (0.6)	6 (1.3)	3 (0.5)	3 (0.5)	0 (0.0)
D. No, because everyone should have the right to choose.	108 (5.9)	27 (4.9)	81 (6.4)	36 (7.7)	36 (5.6)	28 (4.3)	8 (13.8)

Results are presented as *n* (%). Q26 against gender *p* = 0.005, Q26 against year of study *p* < 0.001, Q27 against gender *p* = 0.085, Q27 against year of study *p* < 0.001, *p* for Fisher’s exact test. JD—junior doctors.

**Table 5 ijerph-18-03595-t005:** Attitudes towards seasonal influenza vaccination in Q28 and Q29 (*n* = 1821).

Question Answer	Total (*n* = 1821)	Gender		Year of Study			
		Male (*n* = 555)	Female (*n* = 1259)	1–2 (*n* = 467)	3–4 (*n* = 645)	5–6 (*n* = 651)	JD (*n* = 58)
Q28: What is your opinion about the seasonal flu vaccine?							
A. It is an almost 100% protection against seasonal flu.	126 (6.9)	54 (9.7)	72 (5.7)	32 (6.9)	48 (7.4)	45 (6.9)	1 (1.7)
B. It is not useful because the seasonal flu virus mutates constantly and there is a different type every year.	310 (17.0)	94 (16.9)	214 (17.0)	120 (25.7)	109 (16.9)	75 (11.5)	6 (10.3)
C. It won’t necessary prevent you from contracting the seasonal flu, but the disease will be less serious.	1379 (75.7)	405 (73.0)	969 (77.0)	313 (67.0)	486 (75.3)	529 (81.3)	51 (87.9)
D. Vaccines in general are not effective and the seasonal flu vaccine is not an exception.	6 (0.3)	2 (0.4)	4 (0.3)	2 (0.4)	2 (0.3)	2 (0.3)	0 (0.0)
Q29: How often do you get vaccinated against seasonal flu?							
A. Every other season.	73 (4.0)	23 (4.1)	50 (4.0)	17 (3.6)	21 (3.3)	29 (4.5)	6 (10.3)
B. Every season.	329 (18.1)	96 (17.3)	232 (18.4)	57 (12.2)	88 (13.6)	167 (25.7)	17 (29.3)
C. I have never been vaccinated against seasonal flu.	836 (45.9)	234 (42.2)	599 (47.6)	233 (49.9)	315 (48.8)	271 (41.6)	17 (29.3)
D. I haven only been vaccinated once.	263 (14.4)	80 (14.4)	181 (14.4)	69 (14.8)	108 (16.7)	81 (12.4)	5 (8.6)
E. Not regularly.	320 (17.6)	122 (22.0)	197 (15.6)	91 (19.5)	113 (17.5)	103 (15.8)	13 (22.4)

Results are presented as *n* (%). Q28 against gender *p* = 0.020, Q28 against year of study *p <* 0.001, Q29 against gender *p* = 0.022, Q29 against year of study *p* < 0.001, *p* for Fisher’s exact test. JD—junior doctors.

**Table 6 ijerph-18-03595-t006:** Attitudes towards mandatory vaccination of medical staff and medical students (MS) in Q30 and Q31.

Question Answer	Total (*n* = 1821)	Gender		Year of Study			
		Male (*n* = 555)	Female (*n* = 1259)	1–2 (*n* = 467)	3–4 (*n* = 645)	5–6 (*n* = 651)	JD (*n* = 58)
Q30: Do you think that a vaccine against seasonal flu and hepatitis B should be mandatory for medical staff (attending doctors, nurses etc.)?							
A. No, because everyone should have the right to choose.	227 (12.5)	63 (11.4)	164 (13.0)	75 (16.1)	77 (11.9)	65 (10.0)	10 (17.2)
B. No, because those vaccines are not effective.	21 (1.2)	9 (1.6)	12 (1.0)	9 (1.9)	5 (0.8)	6 (0.9)	1 (1.7)
C. No, because those vaccines are not safe.	7 (0.4)	3 (0.5)	4 (0.3)	3 (0.6)	2 (0.3)	2 (0.3)	0 (0.0)
D. Yes, because medical staff has a greater chance to get infected and then spread the spread the virus.	1566 (86.0)	480 (86.5)	1079 (85.7)	380 (81.4)	561 (87.0)	578 (88.8)	47 (81.0)
Q31: Do you think that a vaccine against seasonal flu and hepatitis B should be mandatory for medical students?							
A. No, because everyone should have the right to choose.	287 (15.8)	85 (15.3)	202 (16.0)	106 (22.7)	96 (14.9)	75 (11.5)	10 (17.2)
B. No, because these vaccines are not safe.	6 (0.3)	2 (0.4)	4 (0.3)	3 (0.6)	1 (0.2)	2 (0.3)	0 (0.0)
C. No, because those vaccines are not effective.	22 (1.2)	8 (1.4)	14 (1.1)	9 (1.9)	6 (0.9)	6 (0.9)	1 (1.7)
D. Yes, because medical students rotate through different departments in a hospital and can spread the virus.	1506 (82.7)	460 (82.9)	1039 (82.5)	349 (74.7)	542 (84.0)	568 (87.3)	47 (81.0)

Results are presented as *n* (%). Q30 against gender *p* = 0.384, Q30 against year of study *p* = 0.038, Q31 against gender *p* = 0.881, Q31 against year of study *p < 0.001*, *p* for Fisher’s exact test. JD—junior doctors.

**Table 7 ijerph-18-03595-t007:** Self-reported coverage for 19 vaccines in country-specific vaccination programs declared by medical students and junior doctors—total sample and against gender.

Types of Vaccination	Vaccine	Total (*n* = 1821)			Gender						*p*
					Male (*n* = 555)			Female (*n* = 1259)			
		Yes	No	Don’t Know	Yes	No	Don’t Know	Yes	No	Don’t Know	
Childhood schedules in all EU/EEA	Tetanus	94.2	3.0	2.7	93.9	3.1	3.1	94.4	3.0	2.6	0.848
	Diphtheria	89.0	3.6	7.4	88.8	3.4	7.7	89.0	3.7	7.3	0.908
	Poliomyelitis	85.7	4.9	9.4	85.4	4.5	10.1	85.9	5.1	9.1	0.704
	Rubella	83.3	6.9	9.8	79.5	8.5	12.1	85.1	6.2	8.7	0.012
	Measles	81.0	8.7	10.3	80.5	9.2	10.3	81.3	8.6	10.2	0.898
	Mumps	80.2	8.6	11.2	77.8	9.5	12.6	81.3	8.3	10.5	0.229
	Pertussis	79.3	8.0	12.7	79.5	8.1	12.4	79.2	7.9	12.9	0.963
	*Haemophilus influenzae* type b	53.8	25.7	20.5	54.2	23.6	22.2	53.7	26.6	19.7	0.287
	Human papillomavirus	25.0	65.4	9.6	7.2	78.4	14.4	33.0	59.7	7.4	<0.001
Additional childhood vaccines	Tuberculosis	60.6	29.8	9.6	57.1	31.7	11.2	62.2	29.0	8.8	0.088
	Hepatitis A	40.5	47.4	12.0	42.9	43.4	13.7	39.7	49.1	11.2	0.058
	Meningococci	32.2	46.8	20.9	33.2	42.0	24.9	32.0	48.8	19.1	0.006
	Pneumococci	25.6	51.8	22.6	26.7	49.9	23.4	25.2	52.6	22.2	0.573
	Chickenpox	24.6	60.6	14.8	23.2	60.2	16.6	25.3	60.8	13.9	0.281
Vaccines recommended both for children and for adults in risk groups	Hepatitis B	83.4	11.3	5.3	78.9	13.3	7.7	85.5	10.2	4.3	0.001
	Influenza/seasonal flu	32.9	62.4	4.6	34.2	58.4	7.4	32.4	64.2	3.4	0.001
Vaccines for adults	Typhus	21.1	56.8	22.1	20.2	54.1	25.8	21.6	58.0	20.4	0.041
	Cholera	12.9	67.4	19.7	14.6	62.9	22.5	12.1	69.5	18.4	0.020
	Smallpox	24.4	53.7	22.0	25.0	50.6	24.3	24.1	55.0	20.9	0.160

Results are presented as %. Yes + No + Don’t know = 100% for every vaccine in every group of respondents. *p* for Fisher’s exact test.

**Table 8 ijerph-18-03595-t008:** Self-reported coverage for 19 vaccines in country-specific vaccination programs declared by medical students and junior doctors—against year of study.

Types of vaccination	Vaccine	Year of study												*p*
		1–2 (*n* = 467)			3–4 (*n* = 645)			5–6 (*n* = 651)			JD (*n* = 58)			
		Yes	No	Don’t know	Yes	No	Don’t know	Yes	No	Don’t know	Yes	No	Don’t know	
Childhood schedules in all EU/EEA	Tetanus	88.0	6.6	5.4	94.7	2.5	2.8	97.7	1.2	1.1	100.0	0.0	0.0	<0.001
	Diphtheria	73.4	8.1	18.4	91.2	2.5	6.4	97.2	1.5	1.2	96.6	3.4	0.0	<0.001
	Poliomyelitis	71.7	8.1	20.1	85.3	5.1	9.6	94.9	2.8	2.3	100.0	0.0	0.0	<0.001
	Rubella	67.2	10.1	22.7	85.4	6.7	7.9	92.3	4.5	3.2	87.9	10.3	1.7	<0.001
	Measles	68.7	12.8	18.4	80.8	8.1	11.2	89.4	6.3	4.3	87.9	10.3	1.7	<0.001
	Mumps	65.7	12.6	21.6	80.0	8.2	11.8	90.3	5.7	4.0	84.5	13.8	1.7	<0.001
	Pertussis	55.5	14.8	29.8	82.3	6.4	11.3	92.6	4.5	2.9	87.9	10.3	1.7	<0.001
	*Haemophilus influenzae* type b	40.3	25.9	33.8	55.8	23.4	20.8	61.9	26.4	11.7	50.0	41.4	8.6	<0.001
	Human papillomavirus	30.4	51.0	18.6	27.6	61.7	10.7	19.0	78.2	2.8	20.7	79.3	0.0	<0.001
Additional childhood vaccines	Tuberculosis	49.0	33.2	17.8	58.1	31.0	10.9	70.7	26.1	3.2	69.0	31.0	0.0	<0.001
	Hepatitis A	52.2	27.8	19.9	40.6	47.3	12.1	33.0	59.9	7.1	29.3	67.2	3.4	<0.001
	Meningococci	38.5	28.7	32.8	31.0	44.2	24.8	29.3	60.2	10.4	27.6	72.4	0.0	<0.001
	Pneumococci	32.5	31.3	36.2	26.5	46.5	27.0	20.0	69.6	10.4	22.4	75.9	1.7	<0.001
	Chickenpox	26.3	51.2	22.5	27.0	56.6	16.4	20.4	70.8	8.8	31.0	67.2	1.7	<0.001
Vaccines recommended both for children and for adults in risk groups	Hepatitis B	79.2	10.1	10.7	84.2	10.5	5.3	85.6	12.7	1.7	84.5	12.1	3.4	<0.001
	Influenza/seasonal flu	33.6	58.0	8.4	28.4	67.6	4.0	35.6	61.4	2.9	48.3	51.7	0.0	<0.001
Vaccines for adults	Typhus	24.2	38.3	37.5	21.9	53.6	24.5	18.6	71.0	10.4	17.2	81.0	1.7	<0.001
	Cholera	18.2	48.2	33.6	14.4	64.3	21.2	8.1	82.0	9.8	5.2	93.1	1.7	<0.001
	Smallpox	29.3	38.3	32.3	26.0	48.2	25.7	19.7	67.9	12.4	19.0	77.6	3.4	<0.001

Results are presented as %. Yes + No + Don’t know = 100% for every vaccine in every group of respondents. *p* for Fisher’s exact test. JD—junior doctors.

## Data Availability

Data is contained within the article and Appendix A. The data presented in this study are available in Appendix A.

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
