# Peer review of "Attitudes and Knowledge of European Medical Students and Early Graduates about Vaccination and Self-Reported Vaccination Coverage—Multinational Cross-Sectional Survey"

_ijerph, 2021, doi:10.3390/ijerph18073595_

Round 1
Reviewer 1 Report
Attitudes and knowledge of European medical students and early graduates about vaccination and self-reported vaccination coverage – multinational cross-sectional survey
The finding of this work is merited to be published. The topic is still relevant and the results of these studies can be used for meta-analysis and review.
The manuscript requires minor modification.
Abstract
line 24: Minor recommendation: One group of students is abbreviated as MS (medical students) and the other as graduates (for junior doctors). It should be better to abbreviate them as JD or GS instead of the abbreviation "graduates". So both abbreviations MS and JD (or GS) will be balanced.
line 29: “Despite being positive towards influenza vaccine …” It should be better written - it is not clear what is meant by positivity.
Materials and Methods
line 76-81: The description of data collection is not entirely clear. Why could there be such a large disproportion of completed questionnaires among students from different countries. On the contrary, what motivated students from Slovenia - the largest proportion of respondents. It is evident that the study was burden by a selection bias. Therefore, it should be explained and described in this chapter.
2.1. Statistical methods
line 90-93: The statistical ANOVA (one-way analysis of variance) method can be applied only for continuous variables, not for categorical ones. Statistical significance was assessed at a significance level of alpha = 0.05. Therefore, it can be assumed that a two-sided 95% confidence interval was applied. - The text should be adapted to used methods and assessments.
Results
Line 98-99: The abbreviation SD - probably the standard deviation - is missing.
Table 1: What does other gender mean? - It should be explained.
Table 2, question 3, answer 4 - Religious beliefs * - no explanation for the asterisk.
Line 129: If the difference in categorical variable between two groups (or possibly among several groups or subgroups) is statistically significant, it should be explicitly stated. In the text of the results, there was very often neglected.
Tables: In general, p-values should be displayed in the tables 2,3,4,5 and 6 so that it could be known where there were statistically significant differences between the assessed groups.
It should be correctly written Haemophilus influenzae type b instead of H.influenzae type B.
LIne 182-185: If 11 respondents mentioned vaccination with non-existent vaccines they should be excluded from final analysis because their questionnaires are burden by information bias. It is also valid for those who reported vaccination against smallpox. The final study results should not be burden by those inaccuracies.
Discussion
Line 197-198: Proportions cannot be extended by a standard deviation. If an interval estimate is required, the confidence interval can be used. In this case, it is possible to report only proportion expressed as a percentage.
Line 294: The vaccination against hepatitis type A is used not only for travel but also at the site of the outbreak. Therefore, it should not be considered as a travel vaccination only.
Limitations
To emphasize strong selection bias and, unfortunately, also information bias (non-existent vaccines, immunization against smallpox more than 10 years after worldwide eradication).
Note: Little disappointment - authors did not focus on finding the relationship between the question answers, for example with the use of logistic regression. If they did, the result of their study would be much more beneficial.
Author Response
Dear Reviewer,
In the beginning, we want to thank you for all valuable remarks. Manuscript has been revised according to your suggestions and we feel this improved its merit significantly. Answers to each comment are provided below.
Authors
Comment 1
The finding of this work is merited to be published. The topic is still relevant and the results of these studies can be used for meta-analysis and review.
We appreciate the time and efforts put into our manuscript and want to thank the reviewer for careful reading and detailed comments.
Comment 2
line 24: Minor recommendation: One group of students is abbreviated as MS (medical students) and the other as graduates (for junior doctors). It should be better to abbreviate them as JD or GS instead of the abbreviation "graduates". So both abbreviations MS and JD (or GS) will be balanced.
We thank the reviewer for this comment. We adapted the abbreviation ‘JD’ for graduates throughout the manuscript.
Comment 3
line 29: “Despite being positive towards influenza vaccine …” It should be better written - it is not clear what is meant by positivity.
We agree that this sentence is not well written. Therefore, we replaced it with the following sentence:
Lines 29-31
‘Even though the potential benefit of the flu vaccination seems to be acknowledged by our participants, only 22.1% of MS and JD declared getting flu shot every or every other season.‘
Comment 4
line 76-81: The description of data collection is not entirely clear. Why could there be such a large disproportion of completed questionnaires among students from different countries. On the contrary, what motivated students from Slovenia - the largest proportion of respondents. It is evident that the study was burden by a selection bias. Therefore, it should be explained and described in this chapter.
We certainly agree with the Reviewer that there is a disproportion in contribution of different countries – with large number of responses coming from Slovenian students and junior doctors. The survey tool was disseminated via channels of the European Medical Students’ Association which included e-mail networks, social media and peer-to-peer messaging. In over-represented countries the level of engagement and motivation of students (Slovenia, Germany, Estonia or Italy) seemed much higher than in others. Despite a number of reminders and additional measures taken to increase the number of responses from under-represented countries, the effects varied. On the other hand, when analysed per-country the characteristics of respondents varied greatly, e.g. in The Netherlands most contributing students came from 1.-2. year, in Hungary from 3.-4. year, whereas in Germany most students were in 5.-6. year. This is the reason why we excluded the idea of data presentation per country. We understand that the aforementioned bias may affect the possibility of drawing general conclusions and made appropriate note in the Methods and Limitations sections.
Methods – Lines 107-111
‘The number of respondents per country varied due to factors such as different ways the survey was disseminated locally and how actively was it promoted by students involved. Additional measures were taken such as sending out reminders or reaching out to individuals from under-represented countries as well as study years within a country with lower participation.’
Limitations – Lines 426-432
‘We observed unbalanced representation of some countries in the study (Figure 1) which may have introduced bias in response analysis and drawing generalised conclusions for whole Europe. For this reason, we did not perform evaluation by country or geo-graphical region but focused on gender and study year of participants. The analysis by nation was also problematic due to disproportion in the study years within given countries. Based on our findings, we know that the level of advancement in medical school correlates strongly with given responses thus making per-country analysis questionable.’
The ‘Materials and Methods’ as well as ‘Limitations’ sections have been modified extensively and we kindly invite the Reviewers to revisit the entire chapter of our resubmitted manuscript.
Comment 5
line 90-93: The statistical ANOVA (one-way analysis of variance) method can be applied only for continuous variables, not for categorical ones. Statistical significance was assessed at a significance level of alpha = 0.05. Therefore, it can be assumed that a two-sided 95% confidence interval was applied. - The text should be adapted to used methods and assessments.
According to this suggestion and consulting with a statistician, we used test for categorical variables, not ANOVA. All the questions in our survey questionnaire were categorical in nominal scale, also gender and years of study because we grouped them as follows: 1.-2., 3.-4., 5.-6. year and junior doctors (JD).
We did not present confidence intervals in a new version of our manuscript, but p-values to compare answers between genders and years of study.
Comment 6
Line 98-99: The abbreviation SD - probably the standard deviation - is missing.
We thank the Reviewer for this comment. We used ‘Standard Deviation’ instead (Line 136)
Comment 7
Table 1: What does other gender mean? - It should be explained.
We added the following explanation to the ‘Methods’ section:
Lines 87-90
‘A closed question on the gender offered three choice options: female, male, or other. If the participants did not identify themselves as either female or male, or did not feel like dis-closing it, they could select ‘other gender’.’
Comment 8
Table 2, question 3, answer 4 - Religious beliefs * - no explanation for the asterisk.
The new version of Tables present data more clearly and the asterisk in ‘Religious beliefs’ has been removed (Table 2, Q3).
Comment 9
Line 129: If the difference in categorical variable between two groups (or possibly among several groups or subgroups) is statistically significant, it should be explicitly stated. In the text of the results, there was very often neglected.
Thank you for indicating this shortcoming with which we certainly agree. We have made extensive changes to the Tables and presentation of data as well as in the text of ‘Results’ section (starting at Line 130). It is now explicitly stated if the difference between categories was statistically significant both the Tables and in the text where appropriate.
Comment 10
Tables: In general, p-values should be displayed in the tables 2,3,4,5 and 6 so that it could be known where there were statistically significant differences between the assessed groups.
New Tables 2, 3, 4, 5, 6 , 7a and 7b include the p values in all categories.
Comment 11
It should be correctly written Haemophilus influenzae type b instead of H.influenzae type B.
The modification throughout the text and Tables has been made to ‘Haemophilus influenzae type b’.
Comment 12
LIne 182-185: If 11 respondents mentioned vaccination with non-existent vaccines they should be excluded from final analysis because their questionnaires are burden by information bias. It is also valid for those who reported vaccination against smallpox. The final study results should not be burden by those inaccuracies.
We agree that these 11 participants should be excluded as the lack of understanding regarding the concept of vaccination can be assumed. We identified and removed those eleven respondents from our database and run the analyses anew. However, approximately 25% of our participants reported having been vaccinated against smallpox and exclusion of this group would result in a major loss of information from other questions. After careful evaluation and discussion with other authors and statisticians, we decided to retain the sample of respondents who answered ‘Yes’ in the question about smallpox vaccination. We elaborated on the potential bias in the Limitations section of the resubmitted manuscript.
Lines 445-450
‘Lastly, about 25% of participants declared immunisation against diseases which are not preventable by vaccination to date or are no longer recommended due to eradication of the pathogen, e. g. hepatitis C or smallpox. Awareness of participant-related factors such as language problems in interpreting specialised medical terms in English, inadequate knowledge about immunisation or a lack of focus in filling the questionnaire oblige us to recommend caution in interpreting the presented data.’
Comment 13
Line 197-198: Proportions cannot be extended by a standard deviation. If an interval estimate is required, the confidence interval can be used. In this case, it is possible to report only proportion expressed as a percentage.
Thank you indicating this inconsistency. Correction in the ‘Discussion’ was made to address this comment.
Line 271-273
‘Significant increase of knowledge on immunisation from second (55%) to fourth year (74%) of medical school was also noticed by Berera and Thompson in American students [25].’
Comment 14
Line 294: The vaccination against hepatitis type A is used not only for travel but also at the site of the outbreak. Therefore, it should not be considered as a travel vaccination only.
We apologize for this inaccuracy. The resubmitted manuscript has been adjusted as follows.
Line 379-382
‘However, especially vaccination against hepatitis A is also highly relevant at sites of outbreaks and for several groups at risk [60]. Underlying reasons for such findings lay outside the scope of our research and would require further investigation.’
Comment 15
To emphasize strong selection bias and, unfortunately, also information bias (non-existent vaccines, immunization against smallpox more than 10 years after worldwide eradication).
We thank the Reviewer for this comment. We fully agree with the points made previously and made the necessary adjustments. The ‘Discussion’ and ‘Limitations’ sections of our manuscript were further elaborated, with an emphasis on selection and information bias.
Discussion – Line 383-389
‘In the open section of our survey, eleven respondents mentioned Lyme disease, hepatitis C, malaria, cytomegalovirus infection, anti-dust allergy and West Nile Fever as diseases against which they were vaccinated. At the point of running the survey, none of those vaccines were available thus those students were excluded from further analyses. Such finding indicate that some MS may have difficulties distinguishing between diseases preventable by vaccines and those which are not and might struggle with the whole concept of vaccination.’
Limitations – starting at Line 425
The ‘Limitations’ sections has been modified extensively and we kindly invite the Reviewers to revisit the entire chapter of our resubmitted manuscript.
Comment 16
Note: Little disappointment - authors did not focus on finding the relationship between the question answers, for example with the use of logistic regression. If they did, the result of their study would be much more beneficial.
Thank you for sharing this comment and suggestion with us. We would like to use logistic regression to find a relationship between the question answers and gender, and years of study. However, it was not possible with our study sample because answers for many questions were selected by very few respondents. In addition, if we crossed them with gender and years of study we obtained very small sample sizes in some cells - sometimes even empty cells (with zero respondents). Therefore we did not present the findings of our study with the use of logistic regression.
Reviewer 2 Report
Abstract
Line 29: being positive towards vaccination – language could be more specific
Line: 32 language to be checked
Introduction
Language needs revising. Sentences are sometimes unclear or inconsistent.
Literature cited refers mostly to non-European countries. It would be more appropriate, given the geographical scope of the present study, to refer to European literature.
Materials and methods
“Messages containing questionnaire and the 76 description of the project were sent out through the EMSA Europe email network to the 77 national and local representatives of the organisation who disseminated it further at their 78 universities” unclear sentence, please rephrase
Questions on socio-demographic characteristics are not listed in the composition of the questionnaire, but reported for data analysis.
Was the survey piloted?
Was the survey developed ex novo or existing tools adapted? How did the authors develop the questions?
How were single-choice response questions structured? Did the authors use e.g. Likert scale for collecting answers?
Results
Figure 1 – the figure would benefit from restyling, e.g. exclude figure title, eliminate horizontal lines, etc). Respondents per country could also be stratified by sex, to provide additional interesting info to the reader
Students in Y1-2 scores less (para 3.1) is the difference statistically significant? This should be stated.
Table 2/3/4/5/6 – The tables needs revision. It is hardly readable and information provided could be synthetized and more effectively displayed
Para 3.2 “the necessity of revaccination”. Perhaps vaccine booster is a more appropriate terminology here. Would be useful if authors could provide more details on vaccination booster for what vaccines.
Para 3.2 Also, it might be useful to distinguish between childhood vaccination programmes and additional vaccinations (e.g. flu) recommended for medical students, and present any relevant results in the text
Para 3.3./3.4 Different proportions of MS are reported as per analysis – it is not clear whether those differences are statistically significant
Para 3.6 – see suggestion above for para 3.2
Discussion
Discussion is well structured and literature is used appropriately. However, authors would need to consider transferability issues when they cite non-EU research. Some ideas for future improvement are interesting
Data were collected in 2016 and this may not reflect current attituted towards vaccination especially in the aftermath of the COVID pandemic. Authors would need to acknowlegde that
Language needs revision
Author Response
Dear Reviewer,
In the beginning, we want to thank you for all valuable remarks. The manuscript has been revised according to your suggestions and we feel this improved its merit significantly. Answers to each comment are provided below.
Authors
Comment 1
Line 29: being positive towards vaccination – language could be more specific
We agree that this sentence was not well written. Therefore, we replaced it with the following:
Lines 29-31
‘Even though the potential benefit of the flu vaccination seems to be acknowledged by our participants, only 22.1% of MS and JD declared getting flu shot every or every other season.’
Comment 2
Line: 32 language to be checked
We thank the Reviewer for this comment. We reformulated the sentence.
Lines 32-33
‘Furthermore, we analysed self-reported vaccination coverage of our participants regarding 19 vaccines.’
Comment 3
Language needs revising. Sentences are sometimes unclear or inconsistent.
We appreciate the time and efforts put into our manuscript and want to thank the Reviewer for careful reading. All authors re-read the manuscript and we adjusted several inaccuracies and inconsistencies throughout the text, not only the ones indicated during revision. They are marked in the resubmitted manuscript in yellow.
Comment 4
Literature cited refers mostly to non-European countries. It would be more appropriate, given the geographical scope of the present study, to refer to European literature.
We thank the Reviewer for this very important point. We did another literature search finding several studies performed in Europe which were now included in the revised ‘References’. We have also acknowledged the fact that not all studies can be translated to European realms where appropriate in the text.
Materials and methods
Comment 5
“Messages containing questionnaire and the 76 description of the project were sent out through the EMSA Europe email network to the 77 national and local representatives of the organisation who disseminated it further at their 78 universities” unclear sentence, please rephrase
Thank you for this comment. We rephrased the sentence in the following way:
Lines 104-106
‘Messages including a description of the project and the questionnaire were sent out through the e-mail network of EMSA Europe. EMSA Europe’s representatives disseminated it further at their local universities.‘
EMSA Europe – European Medical Students‘ Association (the abbreviation was explained earlier the text)
The ‘Materials and methods’ (starting at Line 74) section has been extensively rewritten and developed. We kindly invite the Reviewers to revisit the entire section of the manuscript in order to see if the proposed corrections are suitable.
Comment 6
Questions on socio-demographic characteristics are not listed in the composition of the questionnaire, but reported for data analysis.
Additional information on the construction of the questionnaire was provided in the newly introduced sub-chapter ‘2.1. Questionnaire description’ of the section ‘Materials and Methods’.
Lines 86-90
‘The questionnaire began with open questions on the year of birth, country of enrollment for studies, studied subject and the year of studies. A closed question on the gender offered three choice options: female, male, or other. If the participants did not identify themselves as either female or male, or did not feel like disclosing it, they could select ‘other gender’.’
The study form (Questionnaire) was provided in the Supplementary Materials.
Comment 7
Was the survey piloted?
Yes, the survey was piloted. The following information was added in the sub-chapter ‘2.1. Questionnaire description’ of the section ‘Materials and Methods’:
Lines 93-94
‘The study was piloted and adapted within the research team and the board of EMSA Europe but the responses were not included in the final analyses.’
Comment 8
Was the survey developed ex novo or existing tools adapted? How did the authors develop the questions?
The questionnaire was created ex novo and it was not based on any previously available tool. The following information was added in the sub-chapter ‘2.1. Questionnaire description’ of the section ‘Materials and Methods’:
Lines 90-93
‘The survey was not based on any previously available tool and the questions were created for the research purpose by an international team of medical students involved with the European Medical Students’ Association (EMSA Europe).’
In response to the Reviewer, we would like to provide some background about the development and history of this research project. The survey was developed in the beginning of 2016 by an international team of medical students from the European Medical Students’ Association led by the Medical Science Director of the term 2015/2016 (one of co-authors). After 5 years, the authors decided to come together again and work on publishing data obtained back in 2016. All students involved in the project back then are graduated medical doctors today, completing medical training in different specialties around Europe. All are co-authors of the reviewed article proposal and actively participate in the revisions.
Comment 9
How were single-choice response questions structured? Did the authors use e.g. Likert scale for collecting answers?
The Likert scale was not used for collecting answers, but short-sentence options were given for each question. All questions (Q) were obligatory for completion of the survey except for Q24 about any other vaccines that the participants would like to report on. The following clarification was made in the text.
Lines 78-80
‘Single-choice questions were presented as A-B-C-D where students were asked to select one option. All questions except Q24 (additional vaccination) were obligatory for completion of the form.’
Results
Comment 10
Figure 1 – the figure would benefit from restyling, e.g. exclude figure title, eliminate horizontal lines, etc). Respondents per country could also be stratified by sex, to provide additional interesting info to the reader
We thank the Reviewer for this comment. We redesigned all Tables and the Figure in the resubmitted manuscript.
Comment 11
Students in Y1-2 scores less (para 3.1) is the difference statistically significant? This should be stated.
Thank you for indicating this shortcoming with which we certainly agree. We have made extensive changes to the Tables and presentation of data as well as in the text of ‘Results’ section (starting at Line 130). It is now explicitly stated if the difference between categories was statistically significant both in the Tables and in the text where appropriate.
Comment 12
Table 2/3/4/5/6 – The tables needs revision. It is hardly readable and information provided could be synthetized and more effectively displayed
New Tables 2, 3, 4, 5, 6 , 7a and 7b have been introduced in the manuscript.
Comment 13
Para 3.2 “the necessity of revaccination”. Perhaps vaccine booster is a more appropriate terminology here. Would be useful if authors could provide more details on vaccination booster for what vaccines.
We apologize for the inaccuracy and kindly invite the Reviewer to read the newly adapted text with suggested terminology. In these questions (Q4 and 25) we did not provide any details about specific boosters (like in questions Q5-24 about 19 specific vaccines). The objective here was to find out whether MS and JD were aware in general of the existence of vaccinations for which a booster is needed to maintain protection.
Starting at Line 173
‘3.2. Knowledge and practices regarding booster vaccination’
Comment 14
Para 3.2 Also, it might be useful to distinguish between childhood vaccination programmes and additional vaccinations (e.g. flu) recommended for medical students, and present any relevant results in the text
In questions Q4 and Q25 from sub-section ‘3.2. Knowledge and practices regarding vaccination booster’ we did not break down responses into specific vaccines thus unfortunately could not categorise them as suggested by the Reviewer. The design of the questionnaire did not allow to provide any more details. However, this approach was adopted in the sub-section ‘3.6. Self-reported coverage for 19 selected vaccines’ as further suggested.
Comment 15
Para 3.3./3.4 Different proportions of MS are reported as per analysis – it is not clear whether those differences are statistically significant
Thank you for this suggestion of improvement. In sub-sections 3.3 and 3.4. it is now explicitly stated if the difference between categories was statistically significant both in the Tables and in the text section where appropriate.
Comment 16
Para 3.6 – see suggestion above for para 3.2
Thank you for helping us to improve the presentation of our findings in ‘3.6. Self-reported coverage for 19 selected vaccines’ which we kept in mind when re-designing the Tables. The previous Table 7 was broken into two separate ones: Table 7a for gender analysis and Table 7b for the year of study analysis. Types of vaccines in both Tables are now categorised and the following change was introduced in the text:
Lines 229-233
‘The vaccines were divided into four categories: childhood schedules in all EU/EEA, additional childhood vaccines, vaccines recommended both for children and for adults in risk groups, vaccines for adults. The division was based on recommendations of the European Centre for Disease Control and Prevention (ECDC) and the European Commission.‘
Discussion
Comment 17
Discussion is well structured and literature is used appropriately. However, authors would need to consider transferability issues when they cite non-EU research. Some ideas for future improvement are interesting
We thank the Reviewer for the positive evaluation of our ‘Discussion’ section and question on the transferability when citing non-EU research. We strongly agree with that comment and due to that elaborated further on this issue. We have up-dated literature references, cited more European studies and acknowledged this concern in the text.
Lines 264-265
‘However, geographical and socio-cultural disparities may limit direct comparability of our results [17].’
Lines 288-290
‘Although comparison with groups outside Europe may be biased, similar results were found among 4. year medical students of Lahore Medical & Dental College in Pakistan [31].’
Lines 317-318
‘Recalled studies refer to American and Australian populations therefore transferability remains in question.’
Lines 342-343
‘Such methods would require evaluation in the European populations before drawing solid conclusions.’
Line 359
‘(…) vaccination policies in the US vary from those in Europe.’
Comment 18
Data were collected in 2016 and this may not reflect current attituted towards vaccination especially in the aftermath of the COVID pandemic. Authors would need to acknowlegde that
We added two paragraphs at the end of the ‘Discussion’ section, bringing our results in context with recent developments in the COVID-19 pandemic. New paragraphs at Lines 402-421.
Comment 19
Language needs revision
All authors re-read the manuscript carefully. Several inaccuracies and inconsistencies were adjusted. Thank you very much for bringing our attention to this aspect of our work.